



# Dielectric database of organic Arctic soils (DDOAS)

Igor Savin, Valery Mironov, Konstantin Muzalevskiy, Sergey Fomin, Andrey Karavayskiy, Zdenek Ruzicka

Kirensky Institute of Physics, Krasnoyarsk, 660036, Russia

*Correspondence to*: Muzalevskiy K.V. (rsdkm@ksc.krasn.ru) and Savin I.V. (rsdst@ksc.krasn.ru)

**Abstract.** This article presents a Dielectric database of organic Arctic soils (DDOAS). The DDOAS was created based on dielectric measurements of seven samples of organic-rich soils collected in various parts of the Arctic tundra: Yamal and Taimyr Peninsula, Samoilovsky Island (the Russian Federation), and Northern Slope of Alaska (U.S.). The organic matter content (by weight) of the soil samples presented varied from 35% to 90%. The refractive index (RI) and normalized
attenuation coefficient (NAC) were measured under laboratory conditions by the coaxial waveguide method in the frequency range from ~ 10 MHz to ~ 16 GHz, while the moisture content changed from air-dry to field capacity and the temperature from -40°C to + 25°C. The total number of measured values of the RI and NAC contained in the database is more than 1.5 million values. The created database can serve not only as a source of experimental data for the development of new soil dielectric models for the Arctic tundra but also as a source of training data for artificial intelligence satellite algorithms of
soil moisture retrievals based on neural networks. DDOAS is presented as Excel files. The files of DDOAS are available on http://doi.org/10.5281/zenodo.3819912 (Savin and Mironov, 2020).

## 1 Introduction

The last five-year (2015–2019) and ten-year (2010–2019) averages are the warmest in instrumental records. The global mean air temperature (2m above ground level) for 2019 was around 1.1±0.1°C above the 1850–1900 baselines, used as an
approximation of pre-industrial levels (WMO-No.1248, 2020). Moreover, significant temperature anomalies from +2 to +4 degrees Celsius were observed in the Arctic region. The continuing long-term tendency to increase the average surface air temperature in the Arctic region contributes to the formation of anomalous heat flows deep into the soil, their heating and thawing. If carbon stored belowground is transferred to the atmosphere by a warming-induced acceleration of its decomposition, positive feedback to climate change will occur (Schuur and Abbott, 2011). The field studies that do exist
confirm that permafrost thaw is tightly linked to temperature as well as soil moisture (Davidson and Janssens, 2006). Therefore, remote sensing of the soil moisture plays a key role in determining the rate of soil carbon cycling and carbon emission in Northern environments. Nowadays SMAP and SMOS/MIRAS satellites radiometers operating at a frequency of 1.4 GHz (L-band) (Wigneron et al., 2017), GCOM-W1/AMSR2 satellite radiometer operating at frequencies more than 6.9GHz (Gao et al., 2018), and MetOp/ASCAT satellite radar operating at a frequency of 5.3 GHz (C-band) (Brocca et al.,





2017) are used to the monitoring moisture up to 2.5-5.0 cm topsoil thick (Choudhury et al., 1979; Escorihuela et al., 2010). The permittivity model of soils is an essential element in the physical-based algorithms of soil moisture retrieval with using remote sensing data of the current radiometric satellites. Currently, Mironov's models (Mironov et al., 2009, 2012) of mineral soils used in current SMAP (Walker et al., 2019) and SMOS (Wigneron et al., 2017) physical-based retrievals algorithms. For the reason that, surface horizons of Arctic land cover represents organic-rich soils, the structural

characteristics of which are differing from the ones of mineral soils, the error of soil moisture retrievals in Northern regions is substantially higher than for moderate latitudes (Al-Yaari et al., 2017; Wrona et al., 2017). To date, no one of the known dielectric models of organic soils (Bircher et al., 2016; Jin et al., 2017; Liu et al., 2013; Mironov et al., 2015a, 2015b, 2018, 2020; Mironov and Savin, 2015, 2016, 2019; Park et al., 2019) have been used in operational algorithms of existing satellites to retrieve soil moisture in the Arctic regions. This work presents a unique database of laboratory dielectric measurements of

organic soils samples. These soil samples were taken in various places in the Arctic region. Earlier, based on these soil samples, the dielectric models of organic-rich soils were developed for use in algorithms of soil moisture retrieval in the Arctic region in different frequency ranges (Mironov et al., 2015a, 2015b, 2018, 2020; Mironov and Savin, 2015, 2016, 2019). Taking into account the success of the previously developed and acknowledged dielectric Mironov's model (Mironov et al., 2009, 2012), which was created with including published dielectric measurements in open press (Curtis et al., 1995;

Dobson et al., 1985; Hallikainen et al., 1985), we decided to publish our original high-quality laboratory dielectric measurement data for samples of organic Arctic soils. The created database can serve not only as a source of experimental data for the development of new soil dielectric models for the Arctic tundra but also as a source of training data for artificial intelligence satellite algorithms of soil moisture retrievals based on neural networks (Rodriguez-Fernandez et al., 2015). Moreover as was noted in (Bircher et al., 2016), the temperature-dependent Mironov's dielectric models for organic

substrates should be exploited for use in satellite data applications where negative temperatures are one of the major drivers (e.g., freeze-thaw, permafrost or snow-related products).

## 2 Test sites of soil samples collected

The available soil samples were taken from an organic horizon of soils at four geographically different areas, placed in typical Arctic tundra regions (see Fig. 1). Soil sample No.1 (BV) was collected on the Yamal peninsula not far from the

Bovanenkovo oil and gas field. The landscape of the site was moistened non-drainable tundra. The surface is flat, finely hummocky. Vegetation cover presents sedges, moss with dwarf willow shrubs, projective cover 100%, up to 2-5 cm thick. Soil sample No.2 (MS1) and No.3 (MS2) were also collected on the Yamal peninsula in the area of Marresale weather station on the West coast of the Kara Sea. The landscape of the test site No.2 was moistened tundra with a relatively flat surface. The canopy was presented by a moss-lichen cover with cowberry shrubs, a projective cover of 90-100%, and a

thickness of up to 4-10cm. The topsoil horizon (up to a depth of 5-10 cm) is represented by brown peaty loamy sands. Peat formation decreases rapidly with depth, and deeper the soil is represented by grey sandy loam soils.





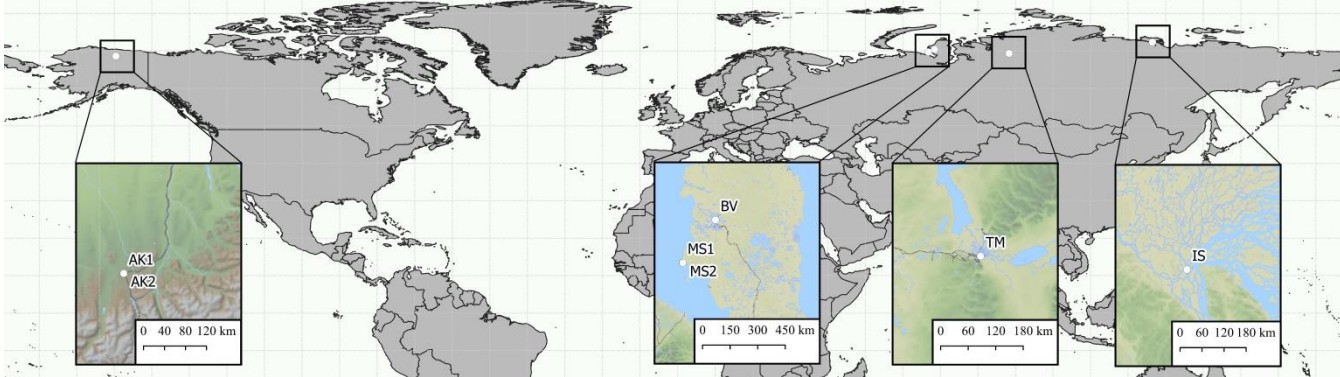

**Figure 1. Location map of soil sampling test sites**


The border between peaty and sandy loam soils is sub-horizontal, gradual, blurred. Place of sampling No. 3 is a peat bog (hillock) with a bumpy-cavity surface. The vegetation canopy was represented by a moss-lichen, cloudberry shrubs, with a projective cover of 100%, thickness up to 3-8 cm, gradually transformation into peat. Peat is brown and slightly decomposed at the surface of the soil. The degree of decomposition of peat increases to medium with increasing depth. Soil samples No.4

(AK1) and No.5 (AK2) were collected from two sites that were on opposite sides of little-used roadway east of Toolik Lake, North Slope of Alaska. The terrain is moist acidic tussock tundra. The landscape has reverted to a dryer condition and is now supporting considerable shrub growth. Place of sampling No.6 (TM) is dry tundra with a relatively flat surface. The vegetation canopy was represented by herbs and mosses, with a projective cover of 90-100% thickness up to 5 cm. Soil sample No.7 (SI) was collected on Samoylov Island at the location of polygonal tundra in the center of a polygon. The

diameter of the polygon was about 12-18 m. The dominant plant was herbal, sedge, dwarf willows. The thickness of the organic layer is 15-25cm; the content of organic matter decreased with soil depths. All soil samples were retrieved from the thawed ground in the form of cylindrical cores 25cm heights and 15cm diameter. The coordinates of the core sampling sites, the depth of soils sampling for dielectric measurements, their dry bulk density and brief mineralogical composition are given in Tab. 1.

**3 Soils samples preparation and method for measuring soils permittivity**

The procedure for measuring and preparing samples is described in detail in (Mironov et al. 2015). At first, the soil was crushed to a homogeneous state in a mortar. Next, it was dried in an oven at 60°C for 24 hours. Then, a certain amount of distilled water was added to dry soil samples of equal volume, after which each sample was thoroughly mixed and sealed for 24 hours to distribute the water inside the sample evenly. The sample thus obtained was placed in a measuring cell, which is

a segment of a coaxial waveguide line with a cross-section of 7/3 mm. The cell length was selected depending on the moisture content of the sample, and, consequently, dielectric loss, 17 mm or 37 mm. The cell volume was 0.529 cm$^3$ and 1.152 cm$^3$, respectively. For uniform compaction of the soil inside the cell, a cylindrical pestle was used.



**Table 1. Sampling points and geophysical characteristics of the studied soil samples**

| № | Site name | Location | Tundra land cover | Depth (cm) | Bulk dry density ($gcm^{-3}$) | Organic matter (%) | Quartz (%) |
|---|---|---|---|---|---|---|---|
| 1 | Bovanenkovo, Yamal Peninsula (BV) | N70.4310, E68.4227 | Mossy-grass | 9-14 | ~0.26 | 50.0 | ~ 30 |
| 2 | Marresale Yamal Peninsula (MS1) | N69.7165, E66.8107 | Mossy-grass | 4-9 | 0.12-0.30 | 61.2 | ~ 25 |
| 3 | Marresale, Yamal Peninsula (MS2) | N69.7152, E66.8180 | Sedge-lichen-sphagnum (polygonal peat, on the rim) | 3-7 | ~0.61 | 34.9 | ~ 40 |
| 4 | East of Toolik Lake North Slope of Alaska, (AK1) | N68.6333, W149.5833 | Shrub (between hillocks) | ~20 | ~0.25 | 80≥ | ~ 8-9 |
| 5 | East of Toolik Lake, North Slope of Alaska (AK2) | N68.6333, W149.5833 | Tussock (top of tussock) | ~20 | ~0.14 | 90≥ | ~ 3 |
| 6 | Taimyr Peninsula (TM) | N69.3523, E88.2832 | Sedge-mossy | 5-7 | ~0.23 | 38.5 | ~ 45 |
| 7 | Samoylov Island (SI) | N72.3697, E126.4834 | Herbal-sedge (polygonal peat, center of the polygon) | 4-7 | 0.23-0.46 | ≤30 | - |

To conduct dielectric measurements, the cell to one of the Rohde & Schwarz ZVK or Keysight PNA-L vector network analyzers was connected, which make it possible to measure the frequency spectra of elements of the scattering matrix $S_{11}$, $S_{22}$, $S_{12}$, and $S_{21}$ in a set frequency range from ~50 MHz to ~15 GHz. An Espec SU-241 heat and cold chamber to change the temperature with a certain step and ensure isothermal measurements with an accuracy of 0.5°C was used. To automate the measurement process, a hardware and software complex consisting of a camera, a network analyzer and a computer

connected via an RS-232 interface or a LAN interface and controlled by a set of built-in commands was developed. This complex made it possible to set the temperature with a specific step and measure the spectra of elements of the scattering matrix as follows. After the temperature control system switched the camera to a predetermined temperature, and this temperature was set inside the camera, control of the standard deviations between the $S_{12}$ spectra, which were measured every minute, began. When the standard deviation decreased to below 0.01, the system recorded all the spectra of the S-

matrix and switched the camera to the next designated temperature point, after which the whole process was repeated. Upon completion of dielectric measurements, the soil sample was removed from the coaxial cell, and its weight moisture and bulk density of the dry soil sample using the thermostatic weighting method was determined. The approximate time of measuring the spectra of the scattering matrix elements for one soil sample, depending on the cell size and moisture of the sample, with a temperature change from -30ºC to 25ºC ranged from 8 to 15 hours.





To obtain the dielectric spectra of soil samples using the measured values of $S_{11}$, $S_{12}$, $S_{22}$, and $S_{21}$, an algorithm developed in (Mironov et al., 2010) was used. This algorithm provides restoration of the real and imaginary parts of the relative complex dielectric constant with errors of less than 9%.

Dielectric measurements were carried out using equipment of the collective use center of the KSC SB RAS. The range of variations in the volumetric moisture, the dry bulk density, the temperature of soil samples and the wave frequency at the

measurements of the refractive index (RI) and normalized attenuation coefficient (NAC) are presented in Table 2.

**Table 2. Range of variations in temperature, moisture, density of soil samples and wave frequency when measuring RI and NAC of soil samples.**

| Test sites (the numbers of soil samples) | Temperature ($^oC$) | Volumetric moisture ($cm^3cm^{-3}$) | Bulk dry density ($gcm^{-3}$) | Frequency (GHz) | Frequency step (GHz) | The total number of measured values |
|---|---|---|---|---|---|---|
| BV (No. 1) | -30…+25 | 0.024-0.428 | 0.715-0.878 | 0.0475-15 | 0.038 | 112000 |
| MS1 (No.2) | -40…+25 | 0.007-0.597 | 0.586-0.772 | 0.015-15 | 0.005 (<1.035GHz) 0.035 (>1.035GHz) | 484800 |
| MS2 (No.3) | | 0.005-0.583 | 0.516-0.689 | 0.015-15 | 0.005 (<1.035GHz) 0.035 (>1.035GHz) | 418140 |
| AK1 (No.4) | | 0.007-0.573 | 0.564-0.665 | 0.01-16 | 0.04 | 177242 |
| AK2 (No.5) | -30…+25 | 0.007-0.599 | 0.498-0.664 | 0.01-16 | 0.04 | 125112 |
| TM (No.6) | | 0.01-0.601 | 0.672-0.855 | 0.015-15 | 0.005 (<1.035GHz) 0.035 (>1.035GHz) | 220584 |
| SI (No.7) | | 0.025-0.593 | 0.917-1.058 | 0.01-15 | 0.038 | 33684 |

**4 Dataset description**

All dielectric and auxiliary measurements were collected at the Dielectric database of organic Arctic soils (DDOAS). The

DDOAS is presented in the files of Excel format (*.xls), and it contains more than 1.5 million measured refractive index (RI) and normalized attenuation coefficient (NAC) (see Table 2). Values of RI, $n$, and NAC, $\kappa$, are related to the value of the complex permittivity, $\varepsilon = \varepsilon' + i\varepsilon''$, where $\varepsilon'$ and $\varepsilon''$ are the real and imaginary parts of complex permittivity, respectively and $i$ is the imaginary unit, by the formulas:

$$n = \sqrt{\frac{\sqrt{(\varepsilon')^2+(\varepsilon'')^2}+\varepsilon'}{2}}, \ \kappa = \sqrt{\frac{\sqrt{(\varepsilon')^2+(\varepsilon'')^2}-\varepsilon'}{2}}. \tag{1}$$

The name of the file corresponds to the name of the test site on which the soil was selected (for example, for test site No.2: MS1.xls). Each file contains a complete set of measured data for the corresponding soil sample: the value of RI and NAC, the wave frequency, the volumetric moisture content, the dry bulk density and the temperature of the soil samples. The variation ranges of these physical values during the measurement for each soil sample are shown in Table 2. The data in each file are organized in the form of tables on separate worksheets (see Tab. 3). The name of each worksheet (tabs) corresponds

to the temperature of the sample at which dielectric measurements were made. DDOAS allows representing the measured values of RI and NAC in three axes: frequency, moisture and temperature dependences.



**Table 3. Presentation of measurement data refractive index and normalized attenuation coefficient on a worksheet of Excel file**

|  | Refractive index | | | | Normalized attenuation coefficient | | |
|---|---|---|---|---|---|---|---|
|  | $W_1$ | $W_2$ | ,..., | $W_M$ | $W_1$ | $W_2$ | ,..., | $W_M$ |
|  | $r_{d1}$ | $r_{d2}$ | ,..., | $r_{dM}$ | $r_{d1}$ | $r_{d2}$ | ,..., | $r_{dM}$ |
| $f_1$ | | | | | | | | |
| $f_2$ | | | | | | | | |
| ,..., | | | | | | | | |
| $f_N$ | | | | | | | | |

$W_j$ is the $j$th value of volumetric soil moisture (cm$^3$cm$^{-3}$), $r_{dj}$ is the $j$th value of soil bulk dry density (g cm$^{-3}$), $f_k$ is the $k$th value of wave frequency (Hz). M and N are the number of individual measurements of soil samples in the range of frequencies from $f_1$ to $f_N$ and moisture from $W_1$ to $W_M$ with correspondent values of soil bulk dry density from $r_{d1}$ to $r_{dM}$.

As an example, Figures 2 and 3 show the frequency spectra of the RI and NAC depending on the volumetric moisture of the soil sample at a temperature of 25°C. In Figures 4 and 5 show the refractive index and normalized attenuation coefficient depending on the temperature of the sample, at a frequency of 1.39 GHz and the different values of volumetric soil moisture. In Figures 6 and 7 show RI and NAC, depending on the volumetric moisture of the samples in the temperature range from -30 to +25 ºC at a frequency of 1.39 GHz.

**5 Data availability**

The DDOAS database is available on Zenodo: http://doi.org/10.5281/zenodo.3819912 (Savin and Mironov, 2020). DDOS data can be reproduced using theoretical permittivity models of Arctic tundra soils, which were early developed based on DDOAS data: spectroscopic (Mironov et al., 2020; Mironov and Savin, 2015, 2016, 2019), and single-frequency (Mironov et al., 2015b, 2018; Savin and Muzalevskiy, 2020) dielectric model.

**6 Conclusions**

This article provides a detailed description of the DDOAS database, which contains more than 1.5 million measured values of the refractive index and normalized refractive index of samples of organic tundra soils taken in various parts of the Arctic region. The DDOAS database can serve as a source of high-quality experimental data on the tundra soils permittivity to develop new dielectric models of the Arctic tundra soils, and can also be used as a training data set for artificial intelligent 145 satellite algorithms of soil moisture retrieval based on neural networks. In the future, the authors plan to continuously supplement the DDOAS database with new dielectric measurements of new soil samples taken from the territories of the Arctic tundra.

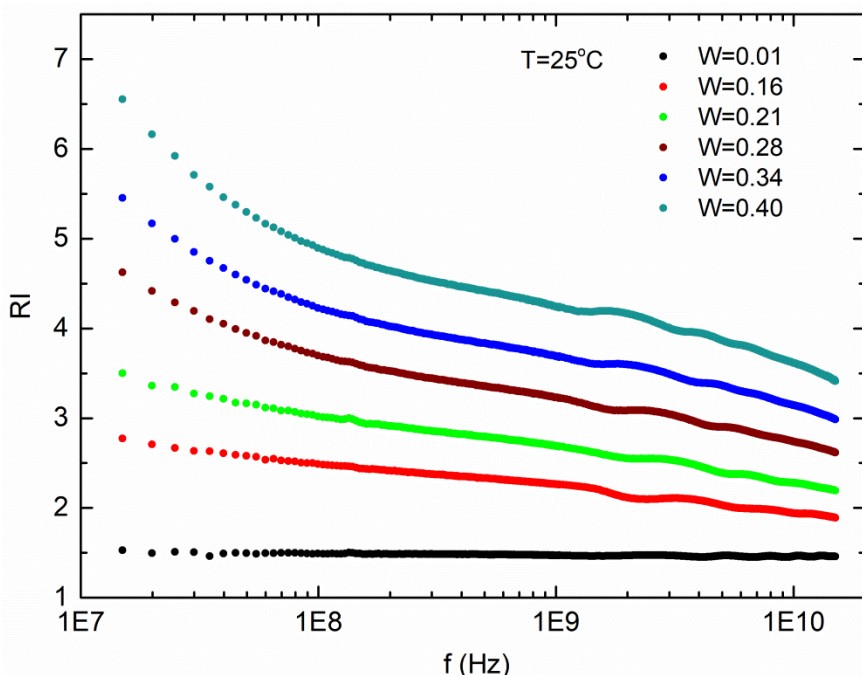

**Figure 2. The spectra of the refractive index (RI) depending on the volumetric moisture content of the soil sample (W) at a temperature of T=25 ° C. Volumetric moisture has dimension W=[cm³cm⁻³] here and in other Figures.**

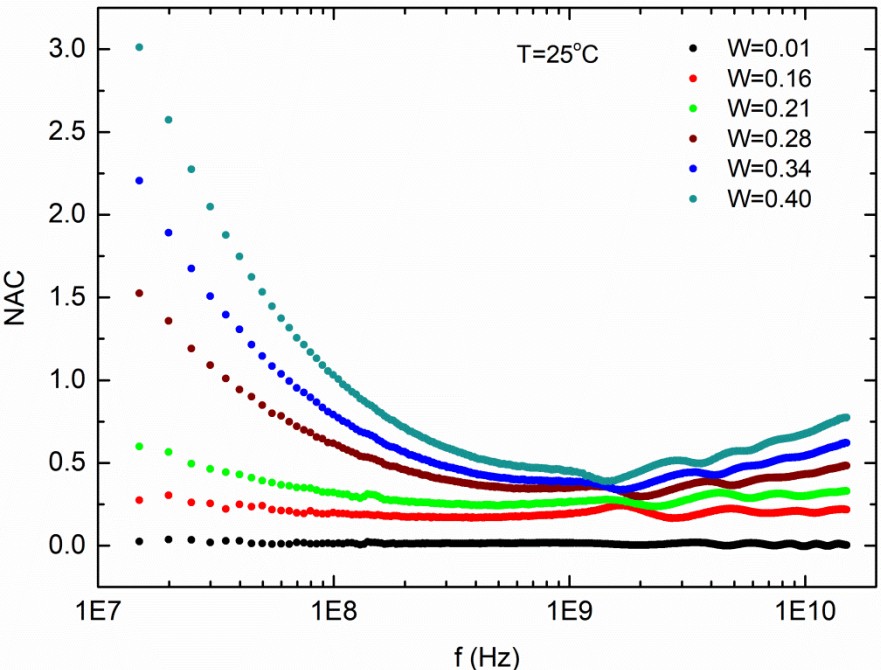

**Figure 3. The spectra of the normalized attenuation coefficient (NAC) depending on the volumetric moisture content of the soil sample (W) at a temperature of T=25 ° C.**



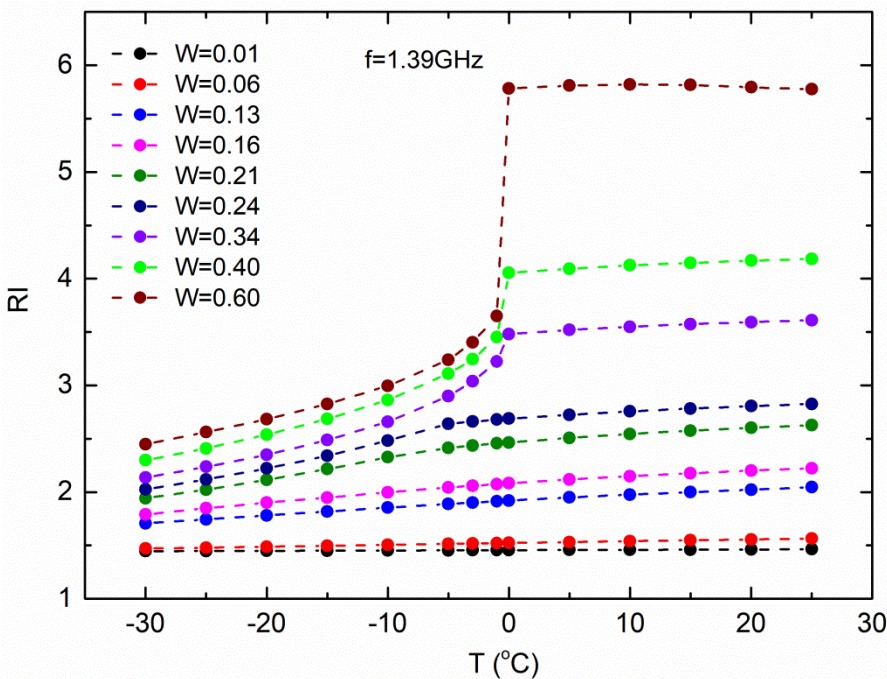

**Figure 4. The refractive index (RI) depending on the temperature of the soil sample (T) for various moisture at a frequency of *f*=1.39 GHz.**


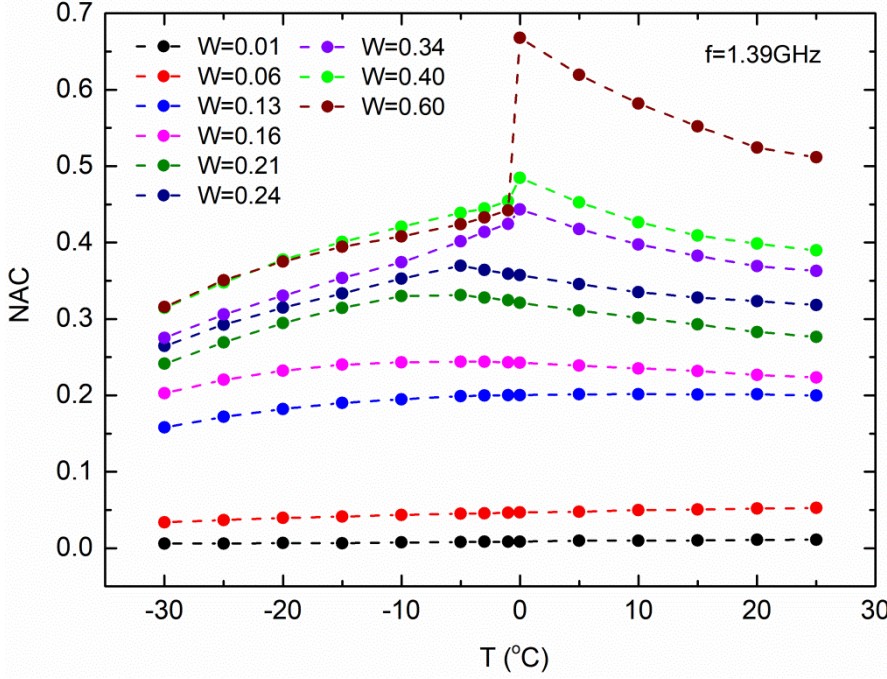

**Figure 5. Normalized attenuation coefficient (NAC) depending on the temperature of the soil sample (T) for various moisture at a frequency of *f*=1.39 GHz.**



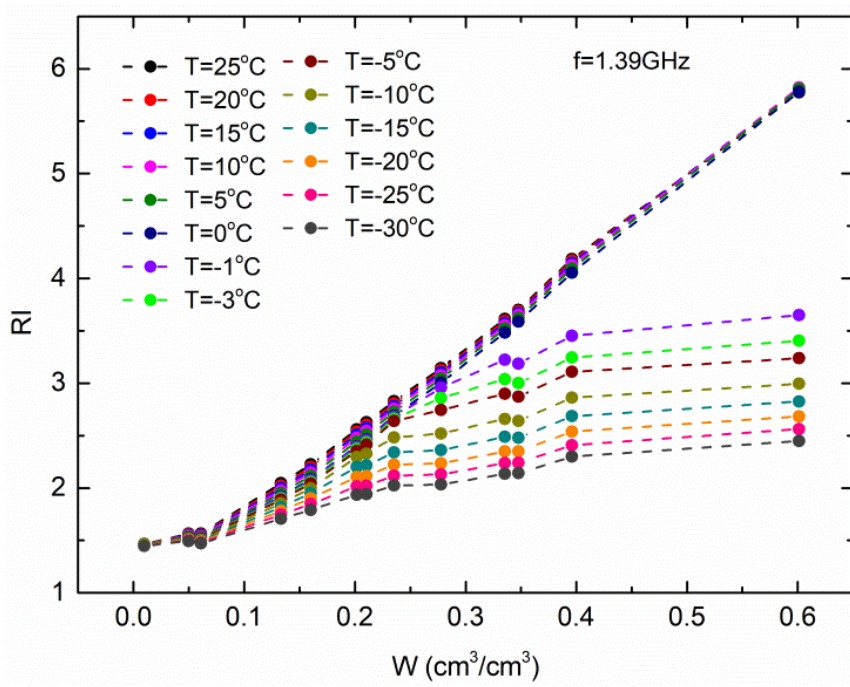

**Figure 6. Refractive index (RI) as a function of soil sample moisture (W) for various temperatures at a frequency of *f*=1.39 GHz.**

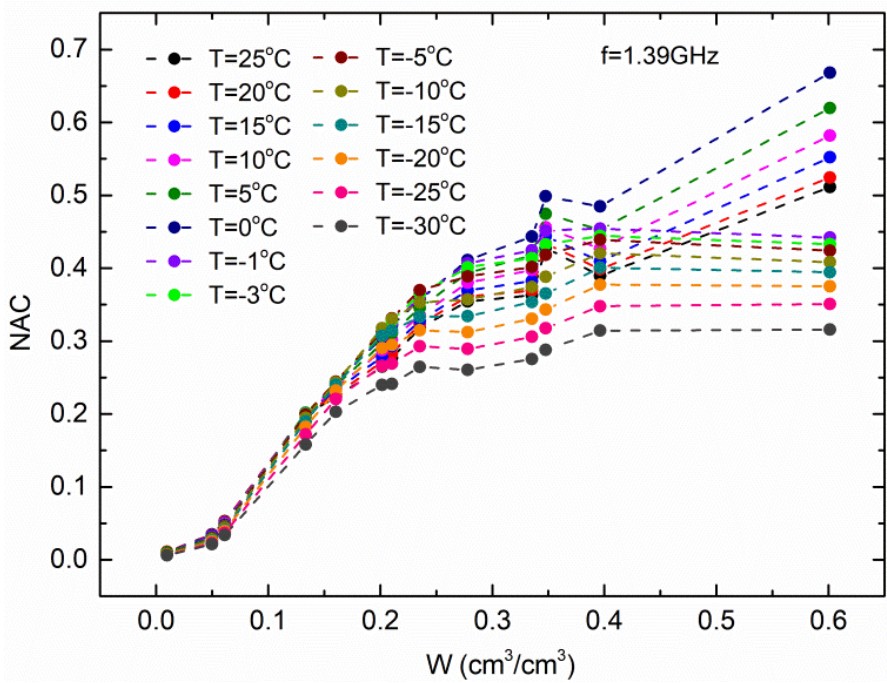

**Figure 7. Normalized attenuation coefficient (NAC) as a function of soil sample moisture (W) for various temperatures at a frequency of *f*=1.39 GHz.**





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
