# Peer review of "Dielectric database of organic Arctic soils (DDOAS)"

_Earth System Science Data, 2020_

## Referee Comment (RC1) · Christopher Simpson (Referee) · 19 Aug 2020

The methods presented in this publication are effective, and the data sets provided are of high quality. The breadth of the data set and the precision with which measurements were made gives the data potential to be instrumental in the development of soil moisture remote sensing solutions for applications in tundra regions. The description of the measurement methodology could benefit from further clarification. I believe there are a couple of typographical errors, with the temperature-controlled chamber being referred to as a "camera" in the publication.

The data is easily accessible. The data set for Samoylov Island (SI) is smaller than the other data sets, leaving readers to speculate about the reason. I think it would be helpful to include a brief explanation for this in the publication. Error estimates and potential sources of error would also help to understand the limitations of the data set.

[Figure]

It would also help to provide a measure of the uncertainty of key parameters.

Regarding the standard deviation measure used in the automation procedure, what are the units of the S12? I think it should read "0.01 dB." Also, how many samples are considered in computing the standard deviation? Is it every value measured since the last temperature change? A moving average? It would be helpful to clarify this, because it gives the reader a better idea of the uncertainty in the measurements.

The data are well-formatted, and they seem to be of high quality. The publication is an appropriate length. It is well-structured, but it could use some clarification in some areas. It reads well in general, but it could benefit further from correcting some grammatical errors and typos.

---

## Referee Comment (RC2) · Anonymous Referee #2 · 25 Aug 2020

This paper presents a dielectric database of organic Arctic soils from four different Arctic regions. Measurements of refractive index and normalized attenuation coefficient were made over a range of frequencies and over a range of soil moisture and temperature. This database has be made available to the community and will be very useful for scientists developing new soil dielectric models and for improving satellite soil moisture retrievals in Arctic regions where soils have a high level of organic matter.

Overall the paper is well written and provides the necessary references. The only thing missing is a comparison and discussion of the different soils. Results are presented for one of the soil samples (MS1), but no comparison is made for the measurements of the other six soil samples. Table 1 gives the properties of the different soil samples, and there should be some discussion of how these different soil properties impact the results shown in figures 2 through 7.

[Figure]

Some minor corrections:

The identifier on the map in Figure 1 "IS" should be changed to "SI".

The captions for figures 2 through 7 should identify the soil sample used to obtain measurements that are presented (MS1).

––––––––––––––––––––––––––––––

---

## Author Response (AR1)

**Response to RC1**

**Comment 1.** I believe there are a couple of typographical errors, with the temperature-controlled chamber being referred to as a "camera" in the publication.

**Answer to comment 1.** In the new version of the manuscript, with regarding you comment, the term "camera" has been changed to the term "temperature chamber" as pointed out in manual to the device.

**Comment 2.** The data set for Samoylov Island (SI) is smaller than the other data sets, leaving readers to speculate about the reason. I think it would be helpful to include a brief explanation for this in the publication.

**Answer to comment 2.** For variuos locations the datasets are differ, due to the fact that they were obtained at different times from 2007 to 2017, and for different purposes, such as creating a model, or testing existing models (as in the case of the Samoilovsky Island soil). That is why the dataset for Samoilovsky Island is much smaller than the others. We intend to expand this database and over time new soils will be added and additional measurements of soils from Samoilovsky Island will be carried out.

For clarification, the following sentence has been added at the Table 2:

"This is due to the fact that the datasets were obtained at different times, from 2007 to 2017 for various purposes and projects, such as creating models, or their testing."

**Comment 3.** Error estimates and potential sources of error would also help to understand the limitations of the data set. It would also help to provide a measure of the uncertainty of key parameters.

**Answer to comment 3.** The estimation of measurement errors is described in detail in (Mironov, V. L., Komarov, S. A., Lukin, Y. I. and Shatov, D. S.: A technique for measuring the frequency spectrum of the complex permittivity of soil, J. Commun. Technol. Electron., 55(12), 1368–1373, 2010; Mironov, V. L., Molostov, I. P., Lukin, Y. I. and Karavaisky, A. Y.: Method of retrieving permittivity from S12 element of the waveguide scattering matrix, in 2013 International Siberian Conference on Control and Communications (SIBCON), pp. 1–3., 2013.).

We have inserted links to these publications at the end of Section 3. We also added additional text for clarifying:

«To obtain the dielectric spectra of soil samples using the measured values of $S_{11}$, $S_{12}$, $S_{22}$, and $S_{21}$, an algorithm developed in (Mironov et al., 2010; Mironov et al., 2013b) was used assuming that only the TEM wave mode propagates in the coaxial cell in the frequency range 0.05–15 GHz. In detail the sources of hardware and measurement method errors describes in the articles (Mironov et al., 2010; Mironov et al., 2013b). »

**Comment 4.** Regarding the standard deviation measure used in the automation procedure, what are the units of the S12? I think it should read "0.01 dB." Also, how many samples are considered in computing the standard deviation? Is it every value measured since the last temperature change? A moving average? It would be helpful to clarify this, because it gives the reader a better idea of the uncertainty in the measurements.

**Answer to comment 4.** Yes, the unit of the standard deviation (SD) of $S_{12}$ is dB. Corrected in text. For SD calculation was used 800 points (evenly distributed in frequency range). Moving average was not implemented (the signal is stable and not noisy). For clarifying measurement process the next text in Section 3 was rewritten:

"After the temperature control system switched the temperature chamber to a predetermined temperature, and this temperature was set inside the temperature chamber, control of the standard deviations between the $S_{12}$ spectra, which were measured every minute, began. When the standard deviation between the current data and the recorded one minute earlier decreased to below 0.01 dB, the system recorded all the spectra of the S-matrix and switched the camera to the next designated temperature point, after which the whole process was repeated."

A new text reads as follows:

"During the measurement, the temperature of the chamber was set by software. After the thermodynamic equilibrium is established in the chamber (monitors by the chamber), the $S_{12}$ value starts to be read every second (to monitor of thermodynamic equilibrium, which establish in the sample). If standard deviation between two successive measurements of $S_{12}$ becomes less than 0.01 dB, then all S parameters measures, and then next temperature in chamber was set and the process was repeated."

**Comment 5.** …it could benefit further from correcting some grammatical errors and typos.
**Answer to comment 5.** Corresponding corrections were made.

**Response to RC2**

**Comment 1.** The only thing missing is a comparison and discussion of the different soils. Results are presented for one of the soil samples (MS1), but no comparison is made for the measurements of the other six soil samples. Table 1 gives the properties of the different soil samples, and there should be some discussion of how these different soil properties impact the results shown in figures 2 through 7.
**Answer to comment 1.** The data in the figures are given for sample No. 6 (TM). Corrected in text and in captions to figures. This sample was chosen as an example, because the obtained spectra are more evenly distributed depend on moisture and they can be represented in the figures completely without overlapping. In addition, similar dependences have not been previously reported for this sample in the literature, in contrast to other samples. This article describes the created database in general, and the given figures rather illustrate the kind of data contained in the database than a comparison of models and dielectric properties of different soils. This is a very large separate work. Comparisons with other samples are not given in this work, since such a comparison in a varying degree in other works was carried out for example (Mironov, V. L., Savin, I. V. and Karavaysky, A. Y.: Dielectric model in the frequency range 0.05 to 15 GHz at temperatures -30°C to 25°C for the samples of organic soils and litter collected in Alaska, Yamal, and Siberian Taiga, in International Geoscience and Remote Sensing Symposium (IGARSS), vol. 2016-Novem., 2016; Mironov, V. L. and Savin, I. V.: Spectroscopic multi-relaxation dielectric model of thawed and frozen arctic soils considering dependence on temperature and organic matter content, Izv. Atmos. Ocean. Phys., 55(9), 986–995, doi:10.31857/S0205-96142019162-73, 2019). In these works, various soil samples are analyzed and compared, which gives an understanding of how certain parameters affect the dielectric properties of soils.

**Comment 2.** The identifier on the map in Figure 1 "IS" should be changed to "SI".
**Answer to comment 2.** Corrected.

**Comment 3.** The captions for figures 2 through 7 should identify the soil sample used to obtain measurements that are presented (MS1).
**Answer to comment 3** Corrections in Figure 1 and captions for Figures 2-7 were made in accordance with the commentary.

**List of changes**

Line 3: Added a co-author Yuriy Lukin

Line 18: "averages" is replaced by "average surface air temperatures"

Line 19: "air temperature" is replaced by "surface air temperature"

Line 28: "more than" is replaced by "above"

Line 30: "to the monitoring moisture up to 2.5-5.0 cm topsoil thick" is replaced by "to monitoring soil moisture in the layer thickness of 2.5-5.0 cm"

Line 38: "have been used" is replaced by "which are using"

Line 44: "which was created with including published dielectric measurements in open press" is replaced by "which was created including the datasets of dielectric measurements published in the open press"

Line 85: "The cell length was selected depending on the moisture content of the sample, and, consequently, dielectric loss, 17 mm or 37 mm" is replaced by "The cells lengths of 17 mm or 37 mm were selected depending on the dielectric loss (moisture content) in the sample"

Lines 90-107: The proposals taking into account the comments of the reviewer have been amended.

"To conduct dielectric measurements, the cell with specimen was placed into the Espec SU-241 temperature chamber and was connected to Rohde & Schwarz ZVK (Keysight PNA-L) vector network analyzer for the measuring of scattering matrix elements: $S_{11}$, $S_{22}$, $S_{12}$, and $S_{21}$. The measurement process was automatized. Temperature chamber and the network analyzer were connected to the computer and controlled by own developed software. This hardware and software complex made it possible to set the chamber temperature (Espec SU-241 accuracy is 0.5°C) with a specific step and measure the spectra of scattering matrix elements. During the measurement, the temperature into the chamber was set by software. After the thermodynamic equilibrium is established in the chamber (monitors by the chamber), the $S_{12}$ value starts to be read every second (to monitor of thermodynamic equilibrium, which establishes in the specimen). If a standard deviation between two successive measurements of $S_{12}$ becomes less than 0.01 dB, then all S parameters measures, and then next temperature in the chamber was set and the process was repeated. These measurements for one specimen takes about 8-15 hours in the temperature range from -30°C to 25°C. As dielectric measurements were finished, the soil specimen was removed from the coaxial cell, its moisture (by weight) and dry bulk density were determined by the thermogravimetric method. To obtain the dielectric spectra of soil specimens using the measured values of $S_{11}$, $S_{12}$, $S_{22}$, and $S_{21}$, the algorithm developed in (Mironov et al., 2010; Mironov et al., 2013b) was used assuming that only the TEM wave mode propagates in the coaxial cell in the frequency range 0.01–16 GHz. In detail, the sources of hardware and measurement method errors describe in the articles (Mironov et al., 2010; Mironov et al., 2013b)."

Line 112: Added a comment to table 2.

"*This is due to the fact that the datasets were obtained at different times, from 2007 to 2017 for various purposes and projects, such as creating models, or their testing."

Lines 132-136: Added clarifications to the data descriptions in Figures 2-7, namely the sample number No.6 (TM).

Lines 150-170: Added clarifications in the captions to Figures 2-7, namely the sample number No.6 (TM).

[revised manuscript text omitted]